# To β or Not to β: How Important Is β-Catenin Dependent and Independent WNT Signaling in CLL?

**DOI:** 10.3390/cancers15010194

**Published:** 2022-12-28

**Authors:** Karol D. Urbanek, Stephan Stilgenbauer, Daniel Mertens

**Affiliations:** 1Mechanisms of Leukemogenesis (B061), German Cancer Research Center, DKFZ, 69120 Heidelberg, Germany; 2Comprehensive Cancer Center Ulm, Ulm University Medical Center, 89081 Ulm, Germany; 3Department of Internal Medicine III, University Hospital Ulm, 89081 Ulm, Germany

**Keywords:** WNT, ROR1, chronic lymphocytic leukemia

## Abstract

**Simple Summary:**

This review aims to present the current state of WNT signaling in chronic lymphocytic leukemia. First, we briefly summarize WNT pathways in physiological conditions and cancer. Then, we focus on WNT pathways in CLL with an emphasis on ROR1. The problem of limited data on canonical WNT signaling in CLL is also highlighted. In conclusion, we suggest that further research should focus on understanding the role of β-catenin signaling in CLL.

**Abstract:**

WNT pathways play an important role in cancer development and progression, but WNT pathways can also inhibit growth in melanoma, prostate, and ovarian cancers. Chronic lymphocytic leukemia (CLL) is known for its overexpression of several WNT ligands and receptors. Canonical WNT signaling is β-catenin-dependent, whereas non-canonical WNT signaling is β-catenin-independent. Research on WNT in CLL focuses mainly on non-canonical signaling due to the high expression of the WNT-5a receptor ROR1. However, it was also shown that mutations in canonical WNT pathway genes can lead to WNT activation in CLL. The focus of this review is β-catenin-independent signaling and β-catenin-dependent signaling within CLL cells and the role of WNT in the leukemic microenvironment. The major role of WNT pathways in CLL pathogenesis also makes WNT a possible therapeutic target, directly or in combination with other drugs.

## 1. Introduction: The Physiological and Pathogenic Role of WNT Signaling

In humans, the WNT family of secreted glycoproteins consists of 19 members [1,2] and 10 dedicated FZD receptors [3]. The term “WNT” comes from the wingless gene found in Drosophila melanogaster and from the proto-oncogene Int-1 (now WNT1) found in mice [1,2]. WNT pathways are traditionally divided into two branches: canonical or β-catenin-dependent WNT signaling and non-canonical or β-catenin-independent WNT signaling. Each of these pathways plays a crucial role in development and homeostasis [1,2,3,4]. However, the division into canonical and non-canonical WNT signaling is artificial and in fact both branches are interconnected [1].

WNT pathways are central to tumor development and progression due to their involvement in proliferation, but usually only one of the two pathways is overactive in each tumor type [1,3]. For example, canonical WNT signaling in cancer is activated by mutations in WNT pathway genes and is crucial for the development of colorectal and breast cancers [1,2]. In contrast, non-canonical WNT-driven tumors such as melanoma and gastric cancer overexpress non-canonical ligand WNT5a and its co-receptor ROR1 [1,3].

Chronic lymphocytic leukemia cells overexpress ROR1, and β-catenin-independent signaling is crucial for CLL development and survival [5,6]. However, several lines of evidence show that the canonical WNT pathway is also active in C2. Activation of WNT signaling is a multi-step process.

Aberrant activation of the canonical WNT pathway is linked to several pathologic entities. The high occurrence of neurological disorders such as schizophrenia, depression or bipolar disorder was linked to the chromosomal translocation of the disrupted in schizophrenia 1 (DISC1) gene, whose protein product inhibited GSK-3β and subsequently activated β-catenin [4]. Other diseases linked to WNT signaling are, e.g., osteoporosis, which is caused by the mutation of LRP5, and fibrosis, where the disruption of the CBP–β-catenin interaction reverses fibrotic injuries in the lungs and kidneys in murine models [4]. Genetic disorders can also be caused by mutations in WNT pathway genes. One example is familial adenomatous polyposis (FAP), a disease where the tumor suppressor gene *APC* is mutated and β-catenin can no longer be degraded. FAP is characterized by the formation of polyps in the large intestine, a pathologic state that can be a precursor of colorectal cancer (CRC) [4].

In fact, *APC* is considered to be one of the most frequently mutated genes in human cancers [4]. The APC protein is a negative regulator of β-catenin, and loss-of-function mutations of APC are responsible in CRC for more than 80% of the aberrant overactivation of the canonical WNT pathway [2]. Interestingly, engineered mutations in the *CTNNB1* gene can lead to the disease progression of CRC, which is identical to what happens when the *APC* gene is mutated, underlining the functional connection of the two genes [2]. Gain-of-function mutations in the *CTNNB1* gene are found in around 30% of cases of hepatocellular carcinoma (HCC). Another gene regulating the WNT pathway, *AXIN1*, accounts for mutations in 15% of cases of HCC, while *APC* is mutated in only 1.6% of HCC cases [2]. However, hyperactivation of the canonical WNT pathway in tumors is not always caused by genetic alterations. Only 10% of lung adenocarcinoma (LUAD) patients carry mutations in WNT genes, but the WNT pathway is active in up to 70% of LUAD cases [2]. This substantial presence of the aberrant activation of WNT in LUAD can be explained by the fact that lung tissue relies on the canonical WNT pathway for regeneration; hence, lung tissue already displays the high expression of WNT ligands [2]. Thus, the type of β-catenin-dependent signaling activation in tumors is tissue-specific and can be driven by the genetic alteration or overexpression of WNT ligands [2]. 

The reliance of cancers on the canonical WNT pathway is not surprising, as it drives proliferation and blocks differentiation [2]. From this perspective, β-catenin signaling can be considered oncogenic, but the role of β-catenin signaling in malignant tumors is more complex. The data from ovarian and prostate cancer, as well as from melanoma, indicate a correlation between high levels of nuclear β-catenin and improved patient outcomes [1]. This correlation of WNT with better outcomes could indicate that the canonical WNT pathway does not always promote proliferation. In fact, stimulation of melanoma cells with WNT3a induced the expression of genes associated with differentiation but suppressed genes with a function associated with proliferation [1]. This differential function of β-catenin shows the complexity of β-catenin-mediated transcription in tumors, and the phenotypic outcome of β-catenin expression may depend on several factors. For example, the WNT coactivators CBP and p300 are closely related, and their sequences are 93% identical, but the two genes regulate different transcriptional programs: CBP regulates proliferation, and p300 regulates differentiation [4]. Β-catenin can also form complexes with several transcription factors, in addition to its classical effectors TCF/LEF. Examples are vitamin A and D receptors as well as androgen receptors [1]. In addition, β-catenin interacts with other pathways—for example, in CRC cell lines with the YAP1 transcriptional complex to induce the expression of the anti-apoptotic *BCL2L1* gene [7].

The involvement of the canonical WNT pathway in the regulation of not only the proliferation but also the apoptosis of tumors makes the WNT pathway an attractive target for therapy. Although there are concerns about the toxicity that arises when inhibiting the WNT pathway, particularly for developing tissues, several compounds and antibodies targeting β-catenin-dependent signaling have entered clinical trials. Strategies to clinically target WNT signaling can be divided into five categories [8]:Inhibition of PORCN to disrupt ligand secretion;Antibodies blocking FZD or ROR receptors;Inhibitors of positive regulators of the WNT pathway, such as RSPO3 or tankyrase;Compounds directly targeting β-catenin and β-catenin-mediated transcription;Activation of the WNT pathway by inhibition of negative regulators; this strategy was developed for tumors such as leukemia, prostate cancer or lymphoma.

Although none of these compounds has yet been approved by the FDA for clinical application, several of these compounds have entered phase II clinical trials. Table 1 summarizes the WNT modulators currently in clinical trials.

## 2. Non-Canonical WNT Signaling Is Dysregulated in Tumors by Differential Gene Expression

Compared to canonical WNT-signaling, much less is known about non-canonical WNT signaling. β-catenin-independent signaling of the non-canonical WNT pathway plays a role in planar cell polarity, tissue patterning, cell migration via modulation of the cytoskeleton, embryonic development and axon guidance, amongst others [2,9]. For these roles, the non-canonical WNT pathway utilizes a different set of WNT ligands and FZD receptors.

WNT ligands that are linked to β-catenin-independent signaling are WNT4, WNT5a/b and WNT11 [9]. Receptors that are linked to β-catenin-independent signaling are FZD_3_, FZD_6_ and FZD_7_ [3,9]. The set of co-receptors is much more diverse as it consists of RAR-related orphan protein 1/2 (ROR1/2), receptor-like tyrosine kinase (RYK) and protein tyrosine kinase 7(CCK4) [3,9]. Due to these different functions, the non-canonical WNT pathways can be further divided into WNT/Planar Cell Polarity signaling (WNT/PCP) and the WNT/Ca^2+^ pathway. The WNT/Ca^2+^ pathway is responsible for the regulation of calcium levels in the cytoplasm, which triggers gene expression and further enhances cellular migration. Upon stimulation of the WNT/Ca^2+^ pathway, the intracellular calcium concentration increases and activates calcineurin, which in turn triggers the expression of NFAT-controlled genes [3]. However, it should be noted that the direct activation of calcium fluxes by WNT stimulation is debatable and there is evidence against it [10]. The WNT/PCP pathway controls cell adhesion and migration via the co-receptors ROR1 and ROR2. ROR1 and ROR2 modulate microtubule stability and actin dynamics through the Rho GTPase and thereby induce the expression of genes responsible for cell adhesion via activation of the JUN-N terminal kinase (JNK) [3]. 

Since the non-canonical WNT pathway is involved in cell migration, it follows that in several types of tumors, dysregulation of non-canonical WNT signaling is associated with metastasis. Overexpression of WNT5a in melanoma, gastric and pancreatic cancer increases metastasis [3,11]. However, β-catenin-independent signaling also stimulates the proliferation of cancer cells. For example, a high level of VANGL-1 (a component of WNT/PCP) is linked to a higher risk of relapse in breast cancer. Similarly, VANGL-2 is responsible for proliferation in breast cancer via the JNK pathway [3]. Other members of the WNT/PCP pathway, such as Celsr1, PRICKLE1, FZD_3_ and FZD_7_, are correlated with poor prognosis in B-cell malignancies [3]. Interestingly, the non-canonical WNT pathway might be involved in the self-renewal of cancer stem cells, since a high level of FZD6 in the mesenchymal stem-cell-like tumor cells of glioblastoma patients correlates with worse prognosis [3]. 

It should be noted that very little is known about mutations of non-canonical WNT pathway genes in cancer. Rather, the activation of non-canonical WNT signaling is achieved by the overexpression of its components, which may make it easier to therapeutically target them. Currently, clinical trials are testing monoclonal antibodies, inhibitors of ROR1 and ROR2 and WNT5a mimetics to modulate non-canonical WNT signaling [8].

## 3. Non-Canonical WNT Signaling Often Inhibits Canonical WNT Signaling

The canonical and non-canonical WNT signaling pathways are often viewed as independent entities sharing only ligands and receptors. This perspective is outdated, and, in fact, both pathways are intertwined. First of all, receptors and ligands are not branch-specific. WNT3a is often viewed as a classic canonical ligand. However, WNT3a can also activate the PLC pathway in the colon cancer RKO cell line [12]. WNT7a was reported to induce proliferation and β-catenin-mediated signaling in ovarian cancer, whereas, in leukemic cells, WNT7a reduces proliferation and has little effect on the activation of β-catenin. This observed tissue-specific difference might be due to the interaction of WNT7a with different FZD receptors. For example, in the presence of FZD_10_, WNT7a stimulation leads to the activation of JNK kinases that are targets of non-canonical signaling [1]. Similar reports can be found on WNT6, WNT9a and FZD_6_ receptor [3,13,14,15]. Both canonical and non-canonical branches of WNT share not only extracellular components but also intracellular components, such as the DVL proteins and the CK1ε enzyme [9]. 

The relationship between CTNNB1-dependent and -independent signaling also includes the interregulation of their activity. It was shown that non-canonical pathways can inhibit canonical WNT signaling on several levels. ROR1 and ROR2 induce the expression of E3 ligase Siah2, which ubiquitinates β-catenin via interaction with APC. This mechanism was reported in hematopoietic stem cells and colon cancer [12,16,17]. Moreover, the WNT/Ca^2+^ pathway can suppress canonical signaling on a transcriptional level by inhibiting the β-catenin complex via NLK kinase, which is activated by calcineurin [3]. These functions of the non-canonical WNT pathway have important consequences for tumor pathology. For example, stimulation of CRC cell lines with a non-canonical ligand, WNT5a, was shown to induce apoptosis [12]. WNT5a can also suppress the growth of ovarian and thyroid cancers [1]. Interestingly, as mentioned above, an increased level of nuclear β-catenin in ovarian cancer is correlated with a favorable patient prognosis, so the growth-suppressing function of WNT5a in this tumor type may not stem from the inhibition of CTNNB1 signaling, but from the regulation via non-canonical WNT signaling. Similarly, the high expression of WNT5a also correlates with improved outcomes in breast cancer [1]. However, knockdown of the receptor ROR1, involved in non-canonical Wnt signaling, in cell lines and xenograft models resulted in growth inhibition, which suggests that ROR1 might be involved in cellular processes besides the non-canonical pathway [1]. On the other hand, high WNT5a expression in melanoma or gastric cancer correlates with poor prognosis [1]. Activation of canonical WNT signaling by the WNT3a ligand or inhibitor of GSK-3β kinase induced apoptosis in melanoma and neuroblastoma cell lines due to the upregulation of the expression of pro-apoptotic gene *BCL2L11* in melanoma, and the downregulation of cyclin D1 in neuroblastoma [18,19,20]. Figure 1 summarizes the interactions between the two branches of WNT signaling.

## 4. WNT Signaling Links Cancer Cells with Their Tumor Microenvironment

Tumor cells often activate WNT pathways in an autocrine manner, which is highlighted by the fact that the inhibition of porcupine—an enzyme required for the posttranslational processing of WNT ligands—results in the apoptosis of cancer cells, pointing to a crucial role of porcupine for tumor survival [2,4,8]. In addition, WNT ligands can also modulate the tumor microenvironment. For example, WNT5a induces the expression of Il12 in dendritic cells, which results in the suppression of myeloid-derived suppressor cells and tumor progression [21]. Activation of canonical WNT signaling in T-cells promotes the protumorigenic Th17 phenotype and drives T-cell exhaustion [2]. Moreover, β-catenin signaling within tumor cells can shape the immune microenvironment around them, as it was shown that melanoma cells with elevated levels of CTNNB1 in the nucleus express IL10 and IL12, which promote immunosuppression. Overall, β-catenin-active tumors respond poorly to anti-CTLA-4/anti-PD-1 immunotherapy [21]. This poor response can limit the usage of GSK-3β inhibitors and other WNT activators in cancer therapy, as they might not be active in combination with specific therapeutics. 

Tumor cells communicate with their microenvironment via several signaling molecules, including WNT. In addition, signaling from the microenvironment can also activate WNT pathways in tumor cells [21]. It was shown that cancer-associated fibroblasts (CAFs) are the main source of WNT2 for CRC [21]. Moreover, WNT2 and WNT5a, secreted by macrophages, play a role in the progression from colorectal adenoma to carcinoma [21]. Activation of canonical WNT signaling in tumors in a paracrine manner is not limited to WNT ligands. Several growth factors and mitogens can also induce CTNNB1 signaling in cancer cells [21]. For example, stimulation with PDGF results in the activation of p68, which in turn inhibits GSK-3β. Similarly, prostaglandin E2 inhibits the destruction complex via the interaction of Gs proteins with Axin [21].

Overall, the modulation of the WNT pathway for cancer therapy may not only be used to inhibit proliferation and/or induce apoptosis in cancer cells but might also be beneficial for immunotherapy.

## 5. Chronic Lymphocytic Leukemia Is an Ideal Model for WNT Signaling and Its Role in the Interaction with the Microenvironment

CLL is the most common leukemia in the western world [22]. It is a disease of mature B-cells and is characterized by patients having more than 5000 CD5^+^CD23^+^ B lymphocytes per ml in peripheral blood. The five-year survival rate for CLL is 87% [23], and this is due to the recent advances in therapy. Standard chemotherapy for CLL has mostly been FCR (NLC) and chemoimmunotherapies that include anti-CD20 mAB [1]. However, novel targeted therapeutics approved for treatment-naïve CLL and/or refractory or relapsed CLL (R/R CLL) are ibrutinib, which inhibits Bruton-agammaglobulinemia-tyrosine-kinase (BTK); idelalisib, which inhibits p110 d isoform-specific phosphoinositide-3 kinase (PI3K); and venetoclax, an inhibitor of anti-apoptotic protein B-cell lymphoma 2 (BCL-2) [22]. 

Both ibrutinib and idelalisib target the B-cell receptor (BCR) signaling pathway, which is central to the pathomechanism of CLL. BCR stimulation with IgM results in the activation of the MAPK-, PI3K- and NF-κB pathways and leads to cell cycle progression, cellular adhesion and the homing of CLL cells to secondary lymphoid organs such as the lymph nodes and bone marrow [24]. Another important pathway for CLL pathogenesis is NOTCH1 signaling. Activation of NOTCH1 leads to the expression of the genes involved in proliferation, survival and metabolism, as well as the *MYC* proto-oncogene [22]. It is important to emphasize the fact that both BCR signaling and the NOTCH1 pathway depend on stimulation from outside of the cell, which highlights the importance of the microenvironment for the development and maintenance of CLL. 

Mutations in both the BCR- and NOTCH1 pathways are of prognostic value in CLL. Immunoglobulin heavy-chain genes (IGHV) constitute the BCR receptor and can be mutated in the development of non-malignant B-cells to increase the affinity of the BCR for antigens. CLL cells with a non-mutated IGHV have a worse prognosis, possibly because they reflect an earlier developmental state [22]. Mutations in NOTCH1 are linked to decreased OS [22]. Another important prognostic factor in CLL is the expression of the ROR1 receptor of non-canonical WNT signaling. ROR1-negative patients have much longer treatment-free survival (TFS) than patients who are ROR1-positive [25]. Other WNT pathway genes are also of prognostic value in CLL. The high expression of WNT3, for example, identifies CLL patients with a favorable prognosis, whereas patients with high levels of the WNT downstream transcription factor LEF1 have lower overall survival [26,27]. The correlation of the functionality of WNT signaling with prognosis in CLL patients suggests the role of WNT signaling in CLL.

## 6. ROR1 Is Central to Non-Canonical Signaling in CLL

ROR1 is expressed at high levels only during the early stages of B-cell development in the bone marrow and is therefore not present on the surfaces of mature B-cells [28,29]. However, in 95% of CLL patients, *ROR1* is highly expressed on CLL cells, induced by STAT3 signaling [25,30]. Moreover, other WNT/PCP genes are often overexpressed in CLL, such as *FZD*_3_, *FZD*_6_, *WNT5b* and *PRICKLE-1* [3,31]. Overexpression of each of these genes also correlates with a shorter TFS [32]. Furthermore, FZD_6_ was shown to be crucial for CLL development in a murine CLL model that overexpressed *TCL1* under the control of the B-cell specific Eµ promoter. *FZD_6_*-null Eµ-TCL1 mice developed less severe leukemia than those with physiological levels of *FZD_6_* [3]. These aberrations suggest the possible importance of the WNT/PCP pathway in CLL.

ROR1 signaling is involved in several processes that are central to the pathomechanism of CLL (Figure 2). Upon stimulation with WNT5a, ROR1 binds and phosphorylates cortactin, which in turn leads to the activation of RhoA and, as a result, enhances the migration of leukemic cells [33]. Another means by which the WNT/PCP pathway regulates mobility in CLL is by the induction of the expression of MMP9, one of the metallopeptidases essential for the dissolution of the extracellular matrix (ECM) [34]. ROR1 also plays a role in the regulation of CLL cell proliferation. ROR1 induces the phosphorylation of ERK1/2 via Rac and the guanine-nucleotide exchange factor DOCK2. ERK1/2 is important for the proliferation of CLL cells, as the knockdown of ERK1/2 slows the division of CLL cells [6]. NF-kB is another pathway activated by ROR1. Activation of NF-kB by ROR1 results in the expression of pro-inflammatory genes *IL-6*, *IL-8*, *CCL2*, *CCL3*, *CCL4* and *CXCL1* [35]. IL-6 activates the STAT3 pathway [35]. Finally, ROR1 is also involved in the inhibition of the canonical WNT pathway, since the knockdown of ROR1 sensitized CLL cells to β-catenin stabilization by WNT3a [36].

WNT/PCP signaling is a highly interconnected network and may interact with other key pathways for CLL. Interestingly, WNT/PCP signaling is connected to the BCR pathway. As mentioned above, BCR is involved in the regulation of cell adhesion and proliferation via the activation of cortactin and ERK1/2 [6,33]. The therapeutic compound ibrutinib can block this activation via the BCR cascade, but ibrutinib has little effect on the activation of WNT induced by ROR1 [6,33]. This redundant activation of ERK1/2 signaling may have major implications for patients treated with ibrutinib and opens the door to combinational therapy with anti-ROR1 ABs and inhibitors. Inhibition of ROR1 signaling may also be an option for patients treated with venetoclax. Recently, it was shown that a high ROR1 level correlates with resistance to venetoclax in CLL, and ROR1 expression increases during treatment with venetoclax. This is most likely due to the activation of the NF-kB pathway by ROR1, which results in the induction of the expression of *BCL2L1*—another anti-apoptotic member of the BCL2 family [37,38]. Due to its involvement in many pathologic processes in CLL, ROR1 is an attractive molecular target for therapy. 

## 7. What Is the Role of the Canonical WNT Pathway in CLL?

The role of β-catenin-dependent signaling in CLL is much more elusive. β-catenin-dependent signaling is undoubtedly active in CLL compared to non-malignant B-cells [39]. Moreover, treatment of leukemic cells with WNT inhibitors induced apoptosis [31]. However, WNT3 stimulation failed to activate β-catenin [26]. Interestingly, CLL cells overexpress several other WNT ligands linked with the canonical WNT pathway: WNT6, WNT9a, WNT10a and WNT16 (Figure 3). However, it should be pointed out that the function of these ligands in CLL has not yet been explored [31]. Along with these WNT ligands, the effector transcription factors LEF1 and TCF4 that are present downstream of WNT signaling are also overexpressed in CLL [27,40]. High expression of the LEF1 TF correlates with an unfavorable OS. This more aggressive CLL phenotype might be due to the repression of the *CYLD* gene, which possibly acts as a tumor suppressor [9]. In the same direction, physiological inhibitors of the canonical WNT pathway, such as DKK1, DKK2 and WIF1, are epigenetically silenced in CLL [41]. These data suggest that not only non-canonical WNT signaling but also the canonical WNT pathway play an important role in CLL, yet the detailed mechanism behind canonical WNT functions in CLL remains to be fully elucidated.

The particular role of canonical WNT signaling in CLL is underscored by genetic mutations: mutations in WNT pathway genes occur in approximately 14% of CLL patients [39]. Mutations affect the extracellular components of WNT signaling (WNT1, WNT10A, DKK2, RSPO4), transmembrane receptors (FZD_5_, RYK), cytoplasmic factors (CSNK1E, PRICKLE1) and nuclear factors that modulate TCF/LEF (CHD8, BRD7, CREBBP, BCL9) [39]. It should be noted that none of these genes is usually mutated in other WNT-driven tumors, with the exception of RSPO4. The impact of these mutations on β-catenin signaling was assessed by luciferase reporter assays in kidney-derived HEK293T cells. Mutations in *CSNK1E*, *WNT1* and *FZD5* resulted in canonical WNT pathway repression, while mutations in *BCL9*, *DKK2* and *RYK* caused canonical WNT pathway activation. Interestingly, BCL9 acts as an enhancer of β-catenin signaling in colon cancer and is considered to be an oncogene [42], and the WT form of BCL9 seems to slightly suppress the canonical pathway. On the other hand, the RYK receptor is usually associated with the non-canonical WNT pathway, but there are also reports that RYK is crucial for the activation of the canonical pathway via DVL proteins in the nervous system [43]. Mutations of canonical WNT signaling impact also the viability of CLL cells. Cells with mutations activating canonical WNT signaling were much more dependent on β-catenin signaling as the knock-down of affected genes resulted in a high apoptosis rate. In contrast, silencing canonical WNT pathway genes with repressive mutations did not affect the viability of the CLL cells significantly [39]. 

Mantle cell lymphoma (MCL) can also help us to understand the role of canonical WNT signaling in hematologic malignancies. The much more aggressive MCL tumors are genetically very similar to CLL, differing mostly in the translocation of Cyclin D1 in MCL, which induces the cell cycle and drives the proliferation of MCL cells. In MCL cell lines, stimulation of the BCR receptor with anti-IgM resulted in the inhibition of GSK-3β and subsequent stabilization of β-catenin within 60 min [44]. Activation of BCR signaling also resulted in the upregulation of WNT16 expression, which maintained β-catenin in the active state even when anti-IgM stimulation was removed. However, in these stimulated cells, β-catenin did not form a complex with TCF/LEF1 TFs but rather with the NF-kB-p65 protein, thereby activating the expression of NF-κB genes. This cross-activation marks an interesting overlap between the canonical and non-canonical WNT pathways to activate NF-κB signaling. It should also be noted that because β-catenin does not activate TCF/LEF1 TFs, the classic β-catenin reporter assay based on TCF4 binding sites to measure WNT signaling activity would fail to show the induction of WNT signaling. Hence, it would be beneficial to determine whether overexpressed WNT ligands in CLL have any effect on the transcriptional landscape of leukemic cells, which is distinct from that of TCF/LEF1 TFs, shedding light on the transcriptome-wide effects of WNT signaling in CLL. 

## 8. WNT Is Part of the Communication of CLL Cells with the Microenvironment

Similar to other malignancies, CLL cells also interact with their microenvironment and this may affect the WNT pathway.

ROR1 is a receptor that is specific to WNT5a; hence, the activation of non-canonical WNT signaling in CLL cells may depend on the WNT5a ligand provided by the microenvironment. The WNT5a ligand can be provided by mesenchymal stromal cells (MSCs) in the bone marrow or by ROR (NLCs) and thereby mobilize CLL cells, as well as contributing to the resistance of CLL to venetoclax [33,38].

The situation is more complex with the canonical WNT pathway. As mentioned above, CLL cells overexpress several canonical ligands, including WNT3. Thus far, attempts to detect the activation of β-catenin by canonical ligands via TCF reporter assays have been unsuccessful [26]. The most likely explanation is that WNT/PCP prevents the activation of the canonical pathway via Siah2 E3 ligase [36]. However, IGHV-Mut CLL patients with higher expression of *WNT3* (WNT3^hi^) have a much better prognosis than WNT3^low^ patients. In addition, decreased expression of *WNT3* is a hallmark of disease progression [26]. This suggests a role of WNT3 in the interaction of CLL with its microenvironment. 

On the other hand, the activation of β-catenin in CLL may not be limited only to WNT ligands [45]. CLL cells co-cultured with stromal cells derived from mouse embryonic livers (BMSC) activated the NOTCH2 pathway in BMSC cells, which resulted in the expression of complement factor C1q in the microenvironmental BMSC cells. C1q activated the LRP5 co-receptor in CLL cells, which inhibited GSK-3β and stabilized β-catenin. β-catenin stabilization was further enhanced by its interaction with N-cadherin, a protein whose expression in CLL is also induced by stromal cells [45].

The WNT pathway’s role in the CLL microenvironment is undoubtedly complex and has positive and negative aspects, which should be explored further.

## 9. The WNT Signaling Cascade Is a Therapeutic Target in CLL

ROR1 plays a major role in the pathophysiology of CLL and is therefore a target for therapy. Indeed, the anti-ROR1 humanized monoclonal antibody cirmtuzumab is currently in several clinical trials for CLL and other B-cell malignancies. 

In a phase 1 clinical trial in 26 progressive or R/R CLL patients, cirmtuzumab was shown to be well tolerated and safe. Most of the adverse events, such as anemia, thrombocytopenia and neutropenia, were directly attributable to CLL. Treatment with cirmtuzumab resulted in a 33% reduction in the surface level of ROR1 and the reduced expression of genes associated with stemness, RhoA and Rac1 pathways. None of the patients met complete or partial response criteria as the treatment lasted only for 4 weeks. However, the median time to next treatment was 262 days. Interestingly, this corresponds with the time at which cirmtuzumab becomes undetectable in plasma [46].

Further clinical trials of cirmtuzumab are focusing on its combination with drugs that are already approved for the treatment of CLL, such as ibrutinib and venetoclax. Since both the B-cell receptor and WNT/PCP pathways regulate the migration and proliferation of CLL cells, and the BTK inhibitor ibrutinib can only block signals from the B-cell receptor, it was reasonable to therapeutically pair ibrutinib with anti-ROR1 mAb. Ongoing clinical trials are testing the combination of ibrutinib and cirmtuzumab vs. ibrutinib alone in treatment-naïve or R/R CLL and R/R MCL patients. The combination of cirmtuzumab and ibrutinib was well tolerated in both diseases and showed superior efficacy over ibrutinib alone. For MCL, the overall response rate (ORR) was 82% and median progression-free survival (mPFS) was not reached after 14.6 months, whereas mPFS for ibrutinib alone was 12.8 months. In CLL, mPFS was also not reached at 14.6 months and the ORR was 91% [47].

WNT-based therapy in CLL is not limited to anti-ROR1 Abs. Currently in development are several small-molecule inhibitors of ROR1: VLS-101, KAN 0439834 and ES425, amongst others [8]. However, there is a risk that these inhibitors may suppress the inhibitory effect of ROR1 signaling on the canonical WNT pathway, which is not the case for cirmtuzumab [36]. Another reasonable possibility to target WNT pathways in CLL is by treatment with PORCN inhibitors, since leukemic cells overexpress several WNT ligands that might be crucial for the pathophysiology of CLL [31,33,34]. Finally, the disruption of β-catenin signaling may also be beneficial for CLL and MCL patients, based on in vitro studies where inhibitors of the canonical pathway activated apoptosis in CLL cells [31,39].

## 10. Conclusions: Next Steps to Address Open Questions concerning WNT Signaling in CLL

There is no doubt that the WNT/PCP pathway plays a major role in CLL. This important role is highlighted by the involvement of WNT/PCP signaling in several key processes in CLL, including adhesion, migration and proliferation, as well as interactions with other pathways, such as BCR, the intrinsic apoptotic pathway and canonical WNT signaling. This involvement makes ROR1 an attractive target for the treatment of CLL, and the anti-ROR1 mAb cirmtuzumab is a promising agent with favorable clinical characteristics. Targeting WNT signaling in CLL patients could open up another targeted therapeutic route for the treatment of CLL.

Canonical WNT signaling in CLL, on the other hand, remains a puzzle. Current data suggest that β-catenin signaling is important in CLL as the treatment of leukemic cells with WNT inhibitors induces apoptosis in CLL cells. However, the exact functions of canonical WNT signaling and its gene targets remain unknown. CLL cells overexpress several canonical WNT ligands, yet, so far, there is no evidence for canonical WNT ligands activating specific pathways within CLL cells. Taking into account the fact that WNT3 expression correlates with the clinical course of the patients, it is, however, very reasonable to assume that CLL cells use this ligand to communicate with their environment. Therefore, future research should focus on finding target cells for WNT3 and other WNT signaling ligands and understanding their role in the pathophysiology of the disease. It should also not be forgotten that although CLL in general depends on WNT/PCP signaling, there might be a subset of patients in whose CLL cells β-catenin signaling is upregulated. An example is CLL cells harboring mutated RYK or BCL9 genes, or, possibly, ROR1neg patients [25]. Such CLL patients could be characterized by a unique pathophysiology. Indeed, a candidate subgroup of patients is those with CLL with trisomy 12. This subtype of CLL patients has a higher likelihood of developing Richter’s syndrome, and when mutations in NOTCH1 co-occur with trisomy 12, the prognosis is very unfavorable [22]. Two canonical WNT pathway genes are located on chromosome 12, WNT1 and LRP6, and their overrepresentation via trisomy 12 could lead to overexpression when compared to non-trisomy-12 cases, thereby activating canonical WNT signaling in CLL with trisomy 12. Recently, it was reported that IGHV-UM trisomy 12 CLL cells showed the enrichment of β-catenin pathway genes [48].

Lastly, it is known that β-catenin can interact with other signaling pathways, such as YAP1 in CRC or NF-κB in MCL, thereby leading to unique transcriptional programs [7,44]. It would therefore be beneficial to identify the binding partners of β-catenin in CLL, as this might reveal a unique set of vulnerabilities in WNT/β-catenin-active CLL patients and offer novel therapeutic options. 

Canonical WNT signaling in CLL is still mostly unexplored and holds many secrets, and therefore it should be a major focus of further research. 

## Figures and Tables

**Figure 1 cancers-15-00194-f001:**
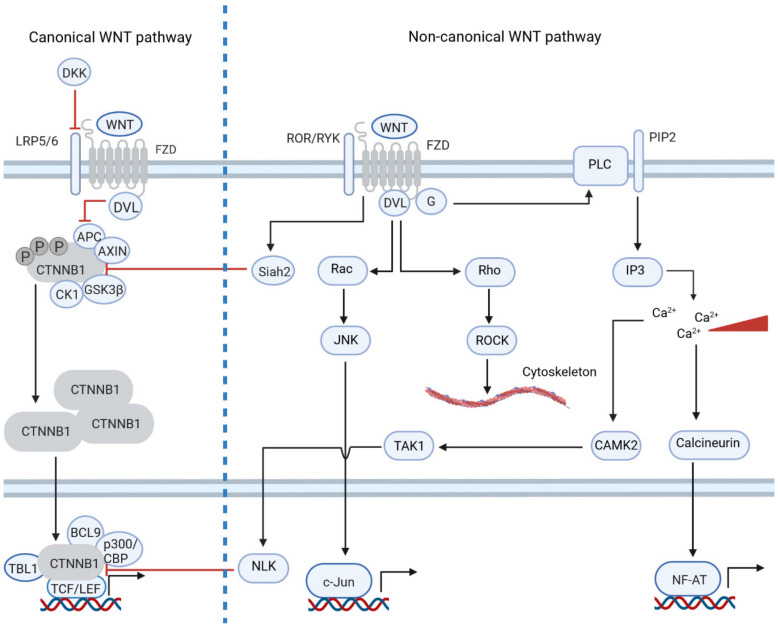
Non-canonical WNT pathways inhibit canonical WNT signaling on several levels. Canonical and non-canonical WNT pathways are usually activated by different sets of WNT ligands and FZD receptors. Upon activation of the canonical pathway, DVL is recruited to FZD and AXIN to LRP5/6, resulting in inhibition of the β-catenin destruction complex. Β-catenin then accumulates in the cytoplasm and enters the nucleus, where it activates TCF/LEF dependent transcription. The non-canonical branch can be divided into the WNT/PCP (planar cell polarity) and WNT/Ca^2+^ pathways. WNT/PCP signaling regulates cell migration via Rho and ROCK kinases and adhesion via JNK kinase. Activation of WNT/Ca^2+^ signaling induces calcineurin, which leads to the transcription of NF-AT genes. Both non-canonical branches can inhibit canonical signaling: WNT/PCP via Siah2 E3 ligase, which induces β-catenin degradation in a GSK-3β-independent manner; WNT/Ca^2+^ via TAK1 and NLK kinase.

**Figure 2 cancers-15-00194-f002:**
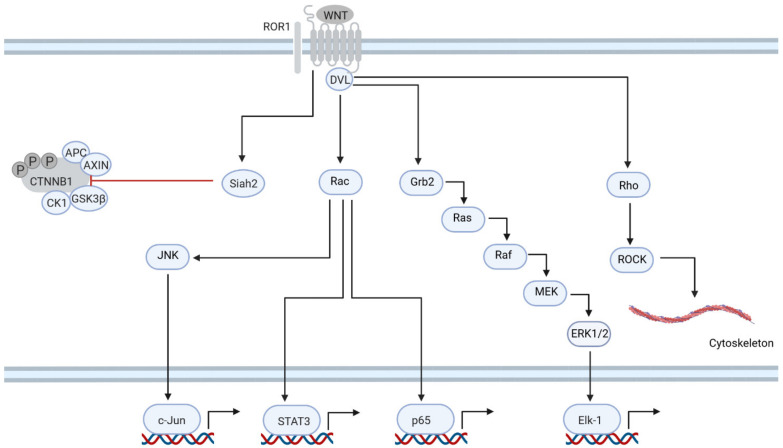
WNT/PCP signaling is specific to CLL and central to the pathomechanism. ROR1 plays a central role in CLL pathogenesis. ROR1 signaling regulates (1) canonical signaling via Siah2 E3 ligase; (2) cell migration via Rho and ROCK kinases; (3) cell adhesion via JNK kinase; (4) proliferation via Ras cascade; (5) NF-kB pathway; (6) STAT3 signaling.

**Figure 3 cancers-15-00194-f003:**
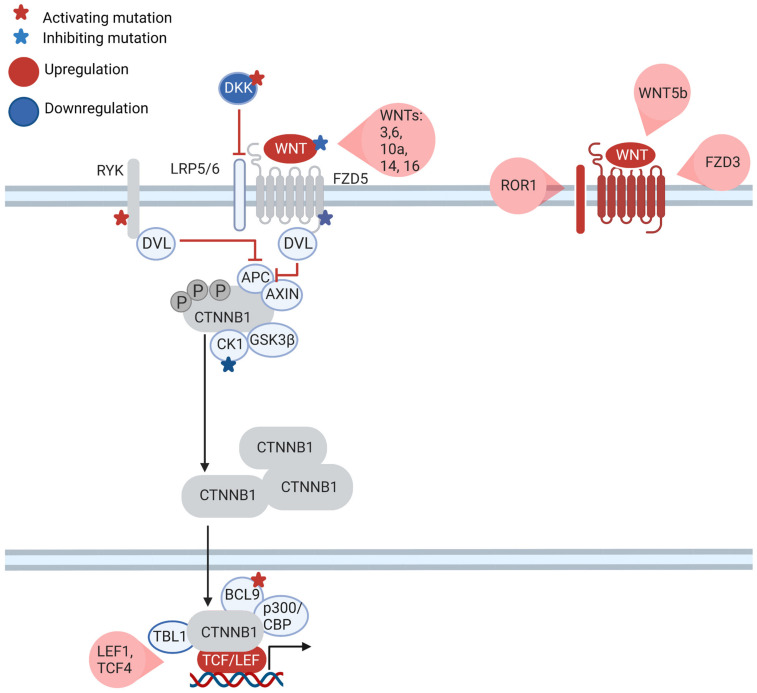
WNT pathway genes are mutated and dysregulated in CLL. Several WNT pathway genes are dysregulated in CLL. CLL cells often overexpress canonical WNT ligands and transcriptional components of canonical branch: LEF1 and TCF4. Pathway inhibitors such as DKK1 and 2 are epigenetically silenced. In non-canonical branch, receptors (FZD_3_, ROR1) and ligands (WNT5b) are overexpressed. Mutations of WNT pathway genes can lead to activation (DKK, BCL9 and RYK) or inhibition (FZD_5_, CK1) of β-catenin signaling.

**Table 1 cancers-15-00194-t001:** WNT pathway modulators in clinical trials. Adapted from [8].

Molecular Target	Function	Drug	Function	Strategy Type	Cancer Type
PORCN	Posttranslational processing of WNT proteins	WNT-C59 WNT974 RXC004 CGX1321 XNW7201 ETC-1922159 IWP compounds	competitive inhibitor competitive inhibitor competitive inhibitor competitive inhibitor competitive inhibitor competitive inhibitor competitive inhibitor	1	Breast Pancreas Colorectal Melanoma Head and neck Cervix Esophageal cancer Lung Gastric Liver
RSPO3	Enhancer of WNT signaling	Rosmantuzumab OMP-131R10	humanized antibody	3	Advanced relapsed tumors and refractory solid tumors
FZD_8_	Receptor	Ipafricept OMP-54F28	truncated decoy receptor	2	Liver Ovary Pancreas
FZD_1/2/5/7/8_	Receptors	Vantictumab OMP-18R5	human IgG2 monoclonal antibody	2	Solid tumors
FZD_10_	Receptors	OTSA101-DTPA- 111ln	radiolabeled monoclonal antibody	2	Sarcoma
TNKS	Positive regulator of the canonical pathway	AZ1366 G007-LK NVP-TNKS656 XAV939 MSC2504877	competitive inhibitor competitive inhibitor competitive inhibitor competitive inhibitor competitive inhibitor	3	Cancer
β-Catenin	Transcription activator	BC2059	protein–protein interaction inhibitor		Desmoid tumor Osteosarcoma AML CML Myelodysplastic syndrome Multiple myeloma Colon Head and neck Liver Melanoma B cell lymphoma Gastric Pancreas
		CGP049090	protein–protein interaction inhibitor		
		CWP232291	β-Catenin degrader		
		MSAB	β-Catenin degrader		
		E7386	protein–protein interaction inhibitor		
		PKF115-584	protein–protein interaction inhibitor		
		PKF118-310	protein–protein interaction inhibitor		
		SAH-BLC9	protein–protein interaction inhibitor	4	
		ICG-001	protein–protein interaction inhibitor		
		PRI-724	protein–protein interaction inhibitor		
		SM08502	competitive inhibitor		
		LF3	protein–protein interaction inhibitor		
DKK-1	Antagonist of WNT ligands	BHQ880 DKN-01	monoclonal Ab monoclonal Ab	5	Liver Biliary tract cancer Gastric Prostate Ovary Multiple myeloma Lung
GSK-3β	Negative regulator of the canonical pathway	LY2090314 9-ING-41	competitive inhibitor competitive inhibitor	5	Leukemia Pancreas Lymphoma Sarcoma Glioblastoma Breast Bladder Kidney Ovary Bone
ROR1	Co-receptor of the non-canonical pathway	Cirmtuzumab ROR1R-CAR-T NBE-002 VLS-101 KAN 0439834 ES425	monoclonal Ab CAR-T therapy Ab–drug conjugate Ab–drug conjugate competitive inhibitor bispecific Ab	2	CLL SLL MCL Breast Lung
ROR2	Co-receptor of the non-canonical pathway	CAB-ROR2-ADC	Ab–drug conjugate	2	Lung Breast Sarcoma
WNT5	Ligand of non-canonical pathway	Foxy-5	WNT5a mimetic	3	Colon Breast Prostate

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
