# Peer review of "To β or Not to β: How Important Is β-Catenin Dependent and Independent WNT Signaling in CLL?"

_cancers, 2022, doi:10.3390/cancers15010194_

Round 1

Reviewer 1 Report

Cancers-2063002 / Review

With regard to the WNT pathway involvement in various cancers, Urbanek KD, Stilgenbauer S and Mertens D provide a novel and exciting overview of the molecular mechanisms underlying WNT activation, dependent or not of beta-catenin, in mouse and human models as well as in treated patients by focusing on Chronic Lymphocytic Leukemia (CLL). 

The introduction describes briefly the main protagonists of the WNT signaling pathways in normal conditions and raises the question of their interconnection or not in pathological models, such as CLL. First, authors depicted the aberrant activation of the canonical beta-catenin-dependent WNT signaling in various disorders and the strategies used in clinic to target this pathway. Second, authors compiled the literature on the malfunctioning non-canonical WNT signaling in several tumors. Then, the authors highlighted the connections between the canonical and non-canonical signaling pathways. Given the importance of the tumor microenvironment, the authors explained how activation of WNT signaling pathways coming from the tumor cells educate their environments and coming from the environments shape tumor cells. Then, the authors proposed that CLL is an interesting pathological model to study the involvement of both branches of the WNT signaling pathways in the dialogue between tumor cells with surrounding cells by arguing that 1) ROR1 is crucial in the non-canonical WNT signaling and contributes to pathomechanism of CLL, 2) Beta-catenin is essential in the canonical WNT signaling despite the lack of detailed mechanism of its function in CD5+ hemopathies and 3) WNT and others molecules that are expressed by stromal cells and NLCs from the microenvironment are significant in the communication with tumor cells. Finally, authors synthetized responses to WNT-based therapy in association or not with other drugs and proposed new therapeutic approaches. The conclusion opens unexplored interesting questions in the field. 

The manuscript is easy to read, carefully written, well-documented with updated and unbiased literature and data are accurately interpreted. This review is illustrated with one table and 3 clear figures. In the field of the WNT signaling in the context of B cells malignancies, the manuscript is a topical issue. At the fundamental level, comparison of the various mechanisms of WNT activation highlights specific and common features of cancers. At the therapeutic level, this review contributes to a new field for the development of more efficient and personalized treatments in CLL. 

To my knowledge, this original manuscript has a strong value-added in the field and should be accepted after minor revisions. 

General comments: The acronyms should be defined and it should be written in the text that the CTNNB1 gene encodes for the beta-catenin protein. Some proteins depicted in the figures, such as TBL1, TAK1 and Elk-1, are never cited/discussed in the main text or in the figure legends. 

 Line 10, Abstract: pahtways should be corrected; 

Lines 18, 19 and 20 Abstract: the last sentence should be rephrased or removed; 

Line 74: b-catenin should be corrected; please check all beta symbols in the text of the manuscript;

- Line 105: Table 2 should be Table 1 as it is written in the text (line 103). It would be helpful for the reader to add in the Table, the 5 strategies to clinically target WNT signaling. Also, FZD1/2/5/7/8 should be written FZD1/2/5/7/8;

Line 223: the sentence should be rephrased;

Page 9, Figure 2: Authors should add another level with the biological responses to help the readers;

Page 11, Figure 3: DKK in the blue circle should be written in white, as WNT in its red circle.  

Line 402: B-cell receptor should be written BCR.

Author Response

Dear Reviewer 1,

First of all, I want to thank you for your elaborate review. It was very kind and helped to improve the manuscript substantially! Below I listed the responses to your comments.

- General comments I added the acronym list at the end of the manuscript. TBL1 protein is mentioned in the table. I also included TAK1 in the Figure 1 description. Elk1 is a member of Ras pathway and was added to the figure to avoid oversimplification.

-  Line 10, Corrected.

- Lines 18, 19, and 20 I removed the sentence.

- Line 74: Thank you for pointing this out. I made sure that β-catenin is spelled correctly in the manuscript.

- Line 105: Table number, as well as subscript of FZD, was corrected. Following your suggestion, I also added a column with the treatment strategy.

- Line 223: rephrased.

- Page 9, Figure 2: The point of the figure was to summarize the signalling network of ROR1 receptor in CLL. We did not add the biological effects of each branch because it would make the figure too complex of filled with text.

- Page 11, Figure 3: corrected.

- Line 402: corrected.

Reviewer 2 Report

This article sets out to review the relative importance of canonical versus non-canonical Wnt signaling in the pathobiology of CLL.  It takes a broad perspective of the data, which is sometimes based on altered expression of individual pathway components, rather than downstream signaling consequences, and attempts to be comprehensive in its approach, despite the existing ambiguities about different mechanisms of non-canonical Wnt signaling and how to measure them downstream.  The emphasis on signaling interactions with the tumor microenvironment is good, but sometimes speculative with regard to WNT signals.  In some cases the authors tend to emphasize the importance of overexpression of a Wnt ligand or receptor in CLL cells without considering whether the respective receptor/ligand is present.  In some cases the overexpression might not be functionally relevant but just a marker or correlate.

Overall, for the general reader there is a danger that they will appreciate the complexity of the problem posed by the article’s title but be uncertain of where the answer lies, i.e. the key arguments in favor of specific pathways may not have sufficient emphasis to rise out of the detailed text as presented. 

In an article about different modes of WNT signaling, it is important to avoid ambiguity in the terminology used, especially in the Abstract.  What is meant by “WNT activation” or the “role of WNT”?  These terms should be more precise and distinguish between ligand and pathway.

There follows number of specific issues, many of which are minor, that will be listed by their line number in the manuscript.  

Line 33.  Delete “interestingly” and change “active to” “overactive”.

43.  Change “this” to “the” (this really is minor!)

44.  The start of this sentence implies general relevance to neurological disorders but the reference cited refers to a single Scottish family, so there’s something wrong here.

50.  For clarity, add “tumor suppressor gene” before APC.

54.  There is no reference for this claim about the frequency of APC mutations.  Do the authors mean in colorectal cancer?  Otherwise this attribute usually goes to TP53 or PI3 Kinase.

59.  Sentence needs a reference.  There are several examples of other sentences needing references in the text.

59 to 61.  Delete this sentence.  It is simply wrong.  Of the 20% or so CRC cases that do NOT have APC mutations, about half of these have activating mutations in beta-catenin.  That these are not found together with APC mutations indicates that there is selection for either one but not both together.

63-64.  “APC is mutated in 1,6% of CRC cases”.  This is clearly a mistake, but I don’t know what was intended.

74.  beta not b

85.  Shouldn’t it be reference 4 instead of 3?

113.  Not sure what is meant here but it is notable that there are components common to both pathways, such as FZD and DVL.

116.  Why are FZDs numbered in subscript throughout?

120-124.  Tread carefully here.  There has been long-standing controversy about the WNT/Ca pathway, and whether WNT proteins directly trigger Ca fluxes.  If the authors know of any paper that demonstrates this (i.e. Ca release as an immediate consequence of cell stimulation by a WNT protein, they should cite it.  Many authors have tried and failed to see such Ca release and some have published their convincing negative results (e.g. Mikels & Nusse 2006, PLoS Biol.). 

130.  Replace “results in” with “is associated with”.

137-139. It’s hard to see how conclusions can be drawn about CSC self-renewal from clinical data about disease prognosis.

149-150.  Conversely, see the above-mention Mikels & Nusse paper showing how Wnt5a can elicit canonical or non-canonical signaling depending on which receptors are overexpressed.  This paper should perhaps be cited in the following paragraph.

176.  Reference missing at end of sentence.

182.  Replace “pathway” with “signaling”

185. Insert “usually” before “activated”.

186-187.  I suggest “Upon activation of the canonical pathway, DVL is recruited to FZD and AXIN to LRP5?6, resulting in inhibition of the beta-catenin destruction complex”.

191.  Recommend deleting “increases the calcium level in the cytoplasm”.

197.  Replace “involved in” with “required for”.

200.  change “is inducing” to “induces”.

210.  I don’t understand this sentence.  It makes ‘WNT’ sound like the only interaction.

211, 212.  References missing at end of sentences.

220.  Insert after “used to”: “inhibit proliferation and/or” and change “can” to “might”.  [The immunotherapy idea is speculative]. 

Starting line 221.  SECTION 6.  It’s odd, and not helpful for the review’s organization, that this section about CLL begins with a description of current therapeutic regimens.  Most readers will first be expecting a summary of the genetics/genomics/transcriptomics, etc., and the % incidence of key driver abnormalities such as NOTCH, TP53, ROR1, etc.  Two recent papers in Nature Genetics (Robbe, P. et al., Nat Genet 54, 1664-74; Knisbacher, B. et al. published online 4th Nov) address this comprehensively and a short summary of their conclusions would be timely.

223. Add “signaling” after “WNT”.

255-256, and the associated paragraph: the authors read too much into the data correlating expression with prognosis, as if mechanistic or cause-and effect relationships can be drawn from the clinical data alone.  

260. “overexpressed” relative to what?  And do those 95% also express Wnt5a/b?  If not, it might not be functional.

262. WNT%B abd PRICKLE have not previously been mentioned.

266. I would say “suggest the possible importance” rather than “highlight the importance”.

273. Replace “transversion” with “dissolution”.  ECM is the standard abbreviation for extracellular matrix.

281.  The title of the reference cited seems to say the opposite of what is stated here.

287.  “inter-connected” would be better.

Starting line 301.  SECTION 8.  Here there should be some indication of the % of CLL to which these statements about overexpression pertain.  It would also help to ask how frequently beta-catenin is stabilized or beta-catenin/TCF target genes are activated, so as to provide evidence of functional activation.  Some of this is discussed in section 11 as next steps.  It’s surprising if there is no existing literature about this.

319-321.  Check grammar in Fig 3 legend.

325-328.  Why are no references given?

332.  It’s unusual to find mutations in WNT1.  Are these common, or maybe just rare passengers?

343.  Reference missing.

362.  “also interact with their microenvironment and this may affect the WNT pathway”.

364.  ROR1 is “a receptor”…, not “the”.

365.  “may depend” on WNT5a provided by…  But it was said previously that CLL cells express WNT5b, whose protein is extremely similar.

376.  I don’t understand the logic of this conclusion.  Decreased WNT3 could just be a marker of progression without contributing functionally.  It could be under the same epigenetic control as other genes, whether linked or unlinked.

384.  Missing Ref.

385.  “The WNT pathway’s role…”

386.  A typo.  “sides, not “sites”.  

390.  Missing Ref.

435-436.  Again, I disagree.  It is pure speculation about the microenvironment at this stage, even though it remains possible.

456.  Better grammar to put “and” before “therefore”.

Author Response

Dear Reviewer 2,

First of all I want to thank you for your elaborate review. It was very informative and helped to improve the manuscript substantially! Below I listed the responses to your comments.

Line 33. Changed.

  1. Corrected.
  2. That was a good point. Because of the full name of the DISC1 gene I wrongly assumed that it was of general relevance to neurological disorders. I rephrased the sentence to stress the fact it was only shown in one family.
  3. Added.
  4. I added the reference and slightly rephrased the sentence to make the statement less strong.

59 Reference added.

59-61. Sentence deleted.

63-64. Thank you for pointing this out. I meant HCC (hepatocellular carcinoma) not CRC.

  1. Corrected (as all other mistakes in β-catenin name).
  2. You are right, reference was corrected.
  3. I meant different WNT and FZD ligands. I clarified it in the text.
  4. To be honest I found this notation in most of the publications.

120-124. I am really grateful for this comment. I was not aware of this controversy. I took a look at some papers and indeed there is no direct evidence that WNT ligands can activate calcium flux. I emphasized that in the review.

  1. Replaced.

137-139. It is a good point, that is why I rephrased the sentence to make it more speculative.

149-150. I am aware that most if not all WNT ligands can activate canonical or non-canonical pathways depending on receptors present on cell surface. But when I started reading about it I realized the complexity of this issue and that it would add unnecessary complexity to the text.

  1. Reference added.
  2. Replaced.
  3. Inserted.

186-187. Thank you for the suggestion. I placed it in the text.

  1. Deleted.
  2. Replaced.
  3. Changed.
  4. I rephrased the sentence.

211-212 – References added.

  1. I followed your suggestion.

Starting line 221.  SECTION 6. We decided to focus on therapeutic options in CLL in this section because later we speculate how WNT pathways and their inhibitors can be used in therapy. Also the review was written for special issue about CLL so we did not want to double information that other authors may write more extensively.

  1. Added.

255-256. I slightly rephrased the sentence to make it more speculative.

  1. I changed it to highly expressed. The appropriate comparison would be normal B-cells.
  2. PRICKLE1 was already mentioned in section 3.
  3. I followed your suggestion.
  4. Replaced.
  5. According to authors, knock-down of ROR1 sensitized CLL cells to WNT3a-mediated β-catenin activation, but inhibition of ROR1 by antibody did not sensitize cells to WNT3a.
  6. Corrected.

Starting line 301.  SECTION 8. This is a good point. Unfortunately, I could not find any literature on this topic apart from differences in WNT genes expression between IGHV UM and M subtypes and normal B-cells. Right now we are working to address some of those questions in our lab.

319-321. Checked.

325-328 References added.

  1. There was only one patient in the cohort with a WNT1 mutation.
  2. Reference added.
  3. Replaced according to your suggestion.
  4. Corrected.
  5. More recent papers are focusing on interactions of WNT5a with ROR1 and did not mention WNT5B, therefore I did not want to speculate about WNT5B.

376 and 435-436. Authors of the publication claim that the WNT3a effect on the clinical course of CLL patients with IGHV-M cannot be explained by any other parameter and so far there was no evidence that WNT3a can activate β-catenin in CLL. Hence we suggested that it might be interesting to explore the role of WNT3a in CLL microenvironment.

  1. Reference added.
  2. Corrected.
  3. Corrected.
  4. Reference added.
  5. Corrected.